# Impact of the COVID-19 pandemic on violence exposure and alcohol use among adults who drink alcohol

**Akua O. Gyamerah**[1,2]*, **Alexandrea E. Dunham**[3], **Janet Ikeda**[3], **Andy C. Canizares**[1], **Willi McFarland**[4], **Erin C. Wilson**[4], **Glenn-Milo Santos**[2,3]

**1** Department of Community Health and Health Behavior, School of Public Health and Health Professions, University at Buffalo, Buffalo, New York, United States of America, **2** Department of Community Health Systems, School of Nursing, University of California, San Francisco, San Francisco, California, United States of America, **3** San Francisco Department of Public Health, Center on Substance Use and Health, San Francisco, San Francisco, California, United States of America, **4** San Francisco Department of Public Health, Center for Public Health Research, San Francisco, San Francisco, California, United States of America

* akuagyam@buffalo.edu

**Data Availability Statement:** The study data has been made available and can be found at the following repository: http://datadryad.org/stash/

## Abstract

The COVID-19 pandemic has exacerbated prevalence of alcohol use and violence, including gender-based violence (GBV); however, little is understood about the pandemic's impact on the relationship between the two. Data were collected from January 2021-April 2023 with adults who drink alcohol (N = 565) in the San Francisco Bay Area. Questions assessed prevalence of heavy alcohol use ($\geq$4 drinks on one occasion $\geq$4 times a month) in the past 3 months and violence/GBV exposure before and during the pandemic. Logistic regression examined associations between violence and alcohol use. Overall, participants reported heavy alcohol use (73.7%), strong desire for alcohol (53.3%), ever experiencing violence (71.6%), and GBV (20.5%). During the pandemic, participants reported experiencing violence (26.1%), more violence than usual (13.8%), GBV (8.9%), and drinking more alcohol (43.7%). People who experienced violence during the pandemic had significantly greater odds of reporting heavy alcohol use (OR = 1.76, p = 0.05) and drinking more during the pandemic than usual (OR = 2.04, p<0.01). Those who reported experiencing more violence during the pandemic than usual had significantly greater odds of reporting heavy alcohol use (OR = 2.32, p = 0.04) and drinking more during the pandemic (OR = 2.23, p<0.01). People who experienced GBV during the pandemic reported a significantly stronger desire for alcohol (OR = 2.44; p = 0.02) than those not exposed. Alcohol-related harms increased over the COVID-19 pandemic, including increased violence/GBV, alcohol use, and an elevated desire to use alcohol among those who experienced violence during the pandemic. Future pandemic preparedness efforts must prioritize violence prevention strategies and adapt alcohol harm reduction, recovery, and treatment programs to pandemic conditions.

share/RoK9itq9yFqVio-7R7M5hqRIL87rsMJ8D-
bEgjjax5M.

**Funding:** This research was supported by funds
from a National Institute of Alcohol Abuse and
Alcoholism (NIAAA) Diversity Supplement
(3R01AA025930-03S1; parent study:
R01AA025930-01A1; PI: Glenn-Milo Santos, PhD).
The funders had no role in study design, data
collection and analysis, decision to publish, or
preparation of the manuscript.

**Competing interests:** None to declare

## Introduction

The COVID-19 pandemic has been globally transformative, superseding a health crisis as social effects began to impact daily life through shelter-in-place policies and quarantine periods. In the San Francisco Bay Area (SFBA), there have been 1.96 million recorded cases of COVID-19 and 10,115 recorded deaths since 2020 [1]. The pandemic has impacted employment, increased social isolation, and worsened conditions for those already vulnerable to health and economic disparities [2,3].

One impact of pandemic-related stressors has been an increase in alcohol use, including increased misuse, relapses, and onset of alcohol use disorder (AUD) [4]. Alcohol sales in the U.S. increased by 20% in the first six months of the pandemic (March 2020 to September 2020) in comparison to sales during the same time period in 2019 [5]. Binge drinking sessions increased by 60% compared to pre-pandemic drinking among binge drinkers and by 28% among non-binge drinkers [6]. Among all drinkers, consumption exceeding the Centers for Disease Control and Prevention's (CDC) drinking guidelines increased by 36% to 38% in the U.S. [7].

Increased alcohol consumption triggered by the pandemic is concerning given the adverse mental and physical health outcomes associated with heavy alcohol use, such as diabetes, sexually transmitted diseases (including HIV), respiratory infections, cancer, depression, cardiovascular diseases, and neuropsychiatric disorders [8]. A cross-sectional study that examined associations between mental health and alcohol use during the pandemic found that those who binge drank during the pandemic had worse mental health outcomes, including higher rates of depression, anxiety, stress, and PTSD in contrast to those who did not binge drink [9]. Other studies have found that heavy alcohol use can worsen the severity of COVID-19 infection [10,11]. Moreover, the number and rate of alcohol-related deaths increased by 25% during the pandemic, outpacing increase in all-cause mortality [12].

In addition to an increase in alcohol use, the pandemic exacerbated the prevalence of violence, including gender-based violence (GBV)—that is, violence targeting a person due to their actual or perceived gender identity or presentation [13–16]. GBV can consist of violence experienced from strangers, the state (i.e., the police, military, etc.), or one's family, community, or intimate partner (IPV). At the start of the pandemic, preliminary reports indicated that there was a rise in GBV globally [13,17]. Studies have since reported observable increases in GBV in various international contexts [18–22], disproportionately affecting women and gender minorities. For example, in a cross sectional study in Kenya and Burkina Faso, there was a significant increase in GBV rates against women during the pandemic despite already high cases [19]. A review of GBV in various African countries similarly found that the pandemic worsened incidence of domestic violence against women. In Argentina, a study among transgender and non-binary people found that 20.5% of trans women, 27.6% of trans men, and 42.5% of nonbinary participants reported an experience of violence during the pandemic [23]. GBV was also observed to worsen in the United States, with a few national and local studies reporting significant increases in violence experiences, especially IPV, among cis women and gender minorities [24,25].

While the impact of the pandemic on alcohol use and on GBV has been documented, few studies have examined the co-occurrence between the two—an issue of concern given prior research demonstrating a relationship between alcohol use and violence [26]. Previous studies have shown that there is an association between alcohol availability and increases in violence as a result of increased isolation and income loss—two related stressors that were common during the COVID-19 pandemic [27–29]. In the United States, a study using 911 calls in Michigan documented that the strength of the relationship between domestic violence reports and

alcohol sales more than doubled since the start of the pandemic [30]. Another U.S. study that examined COVID-19-related stressors and IPV on alcohol use among adults found increased IPV experiences were associated with higher levels of alcohol consumption [31]. While there is growing literature on alcohol use and violence during the pandemic [30–33], there are few studies that specifically look at the relationship between substance use and GBV, rather than just IPV and domestic violence.

The current study thus aimed to examine the relationship between the impact of lifetime exposures of violence (sexual, physical, and verbal), including GBV, on alcohol use and the impact of the COVID-19 pandemic on violence exposure and alcohol use among adults living in the SFBA.

## Materials and methods

### Study design and setting

The present study was a supplemental study administered during the pre-screening for a randomized control trial testing the efficacy of kudzu extract, an herbal supplement, in reducing alcohol use among adults with alcohol use disorder (AUD) (clinicaltrials.gov #: NCT03709043). We analyzed cross-sectional quantitative pre-screening data on alcohol use, violence experiences, and COVID-19 impact among alcohol-using adults (N = 565). The study was conducted at the San Francisco Department of Public Health in the San Francisco Bay Area (SFBA), which was among the first cities to implement a shelter-in-place ordinance at the start of the COVID-19 pandemic [34].

### Eligibility criteria and procedures

Data collection occurred from January 2021 to March 2023 in SFBA. Prospective participants were recruited through street outreach, recruitment flyers, sexual health clinics, needle exchanges, community organizations, bars, websites, and social media. To ensure multiple modes of contact, interested parties could contact our recruitment team by phone or email. To be eligible for this study, participants had to be age 18 years or older, a resident of SFBA, sexually active, and drink alcohol, which was different from the criteria for the clinical trial. Participants were screened by phone by trained study recruiters. Participants were briefed on study procedures and their rights as study participants and provided informed consent before proceeding to the screening questions. All participants provided verbal informed consent.

### Ethical approval

Study procedures were approved by the University of California, San Francisco Committee on Human Research (18–25073).

### Measures

**Demographic variables.** Questions assessed sociodemographic background (age, gender, race/ethnicity) and HIV status. Race and ethnicity were defined using the U.S. government definitions [35] and were categorized as Non-Hispanic/Latino Black, Asian/Hawaiian/Pacific Islander, and Mixed/Other, White, and Latino/Hispanic. We combined non-Hispanic/Latino Native American, mixed, and other racial categories under "Other". The gender categories were cis male, cis female, queer/non-binary/other, and trans female, and trans male.

**Independent variables.** Five types of violence exposures were assessed for their impact on alcohol use before and during the pandemic: 1) violence ever ("In your lifetime, have you ever been verbally threatened, slapped, punched, kicked, beaten up, or otherwise verbally,

physically, or sexually hurt by someone, whether a romantic partner, friend, family, or stranger?"); 2) GBV ever (Do you think any of these [lifetime] experiences of violence happened because of your gender identity or presentation?), 3) violence during the pandemic ("During COVID-19 shelter-in-place order, have you ever been verbally threatened, slapped, punched, kicked, beaten up, or otherwise verbally, physically, or sexually hurt by someone, whether a romantic partner, friend, family, or stranger?"); 4) more violence during the pandemic (assessed as yes/no to: "During the COVID-19 pandemic, I have experienced more verbal, physical, or sexual violence than I usually do"); and 5) GBV during pandemic (assessed as yes/no to: "Do you think any of these [COVID-19] experiences of violence happened because of your gender identity or presentation?").

**Dependent variables.** The outcome variables of interest were: 1) heavy alcohol use (defined as $\geq$ 4 drinks on one occasion $\geq$ 4 times a month) in the previous 3 months (assessed as yes/no); 2) strong desire for alcohol (assessed as yes/no to: "In the last 30 days, have you had a strong desire to drink that made it difficult to think about anything else and that often resulted in the onset of drinking?"); and 3) drink more alcohol during the COVID-19 pandemic and changes related to it like shelter in place orders, changes to social life, employment, etc. (assessed as yes/no). Our definition of heavy alcohol use was derived from the Substance Abuse and Mental Health Services Administration's (SAMHSA) definition of heavy alcohol use as binge drinking (5 or more alcoholic drinks for males or 4 or more alcoholic drinks for females on the same occasion) on 5 or more days in the past month [36].

## Data analysis

Data analysis was conducted using Stata 17 [37]. Bivariate logistic regression analyses examined bivariate associations between violence exposures and alcohol use outcomes. In multivariate logistic analysis, we examined the associations between alcohol use and violence exposures, controlling for gender, age, race and ethnicity, and HIV status. Statistical significance was set at P-value $\leq$0.05.

## Results

### Participant characteristics

There was a total of 565 participants in the present study. Mean age was 41.6 years (standard deviation [SD] = 13.3) (Table 1). By gender identity, 49.7% identified as cis men, 43.5% as cis women, 4.1% as queer/non-binary, 1.8% as trans women, and 0.9% as trans men. A plurality identified as White (49.6%), followed by Latino/Hispanic (23.0%), Black (11.6%), Asian/Hawaiian/Pacific Islander (9.1%), and other (Native American, mixed, and other).

### Alcohol use

Overall, 73.7% of participants reported heavy alcohol use ($\geq$4 drinks on one occasion $\geq$4 times a month) and 52.2% reported a strong desire for alcohol in the past month. Close to half of participants (47.2%) reported drinking more alcohol during the pandemic compared to before, 41.8% reported drinking less alcohol, and a minority (11.0%) reported drinking the same amount (Table 1).

### Violence exposure before and during COVID-19 pandemic

Overall, 72.5% of participants reported experiencing violence and 23.1% reported GBV in their lifetime (Table 1). Over one quarter (27.9%) of participants reported experiencing sexual, physical, or verbal violence during the pandemic and 15.6% reported experiencing more

**Table 1. Demographics, alcohol use, and violence exposure among SFBA adults (N = 565).**

| Variable | N (%)* |
|---|---|
| **Age** | Mean = 41.6 years; SD = 13.3 |
| **Gender** | |
| Cis man | 280 (49.7) |
| Cis woman | 245 (43.5) |
| Queer/non-binary | 23 (4.1) |
| Trans woman | 10 (1.8) |
| Trans man | 5 (0.9) |
| **Race/Ethnicity** | |
| Black** | 64 (11.6) |
| Latino/Hispanic | 127 (23.0) |
| Asian/Hawaiian/Pacific Islander** | 50 (9.1) |
| Mixed/Other** | 37 (6.7) |
| White** | 274 (49.6) |
| **HIV status** | |
| Not living with HIV | 495 (89.8) |
| Living with HIV | 56 (10.2) |
| **Violence experiences** | |
| Violence ever | 391 (72.5) |
| Violence during pandemic | 152 (27.9) |
| More violence during pandemic | 86 (15.6) |
| GBV ever (n = 515) | 119 (23.1) |
| GBV during pandemic | 60 (10.7) |
| **Alcohol use** | |
| Heavy alcohol use ($\geq$4 drinks on one occasion $\geq$4 times a month in past 3 months) | 409 (73.7) |
| Strong desire for alcohol $\leq$ 30 days | 289 (52.2) |
| **Alcohol use during COVID-19** (n = 509) | |
| Drank More | 240 (47.2) |
| Drank Less | 213 (41.8) |
| Drank Same | 56 (11.0) |

*Percentages expressed of those with non-missing responses, categories may not add up to total.
**Non-Hispanic/Latino.

violence than usual during the pandemic. Additionally, 10.7% reported experiencing GBV during the pandemic.

## Bivariate associations between violence and alcohol use

Table 2 reports bivariate associations of violence exposure and alcohol use. Participants who reported a lifetime experience of violence had greater odds of reporting a strong desire for alcohol in the past month (OR = 1.67; 95% CI: 1.1–2.4; p = 0.01). Similarly, those who reported a lifetime experience of GBV had greater odds of reporting a strong desire for alcohol in the past month (OR = 1.53; 95% CI: 1.0–2.3; p = 0.04).

Participants who reported experiencing violence during the COVID-19 pandemic had greater odds of reporting heavy alcohol use (OR = 1.79, 95% CI: 1.1–2.9; p = 0.01) in the past month, a strong desire for alcohol (OR = 2.51; 95% CI: 1.7–3.7; p<0.001) in past month, and drinking more during the pandemic (OR = 1.63, 95% CI: 1.1–2.4; p = 0.01).

Table 2. Bivariate associations of violence exposure and alcohol use (N = 565).

| Violence exposure | Heavy alcohol use OR [95% CI] | P-value | Strong desire for alcohol OR [95% CI] | P-value | Drink more during pandemic OR [95% CI] | P-value |
|---|---|---|---|---|---|---|
| Violence ever | 1.30 [0.9–2.0] | 0.22 | 1.67 [1.1–2.4] | *0.01 | 1.14 [0.8–1.7] | 0.49 |
| GBV ever | 0.88 [0.6–1.4] | 0.58 | 1.53 [1.0–2.3] | *0.04 | 1.19 [0.8–1.8] | 0.42 |
| Violence during pandemic | 1.79 [1.1–2.9] | *0.01 | 2.51 [1.7–3.7] | *<0.001 | 1.63 [1.1–2.4] | *0.01 |
| More violence during pandemic | 2.07 [1.1–3.8] | *0.02 | 2.19 [1.3–3.6] | *0.002 | 2.02 [1.2–3.3] | *0.004 |
| GBV during pandemic | 0.75 [0.4–1.4] | 0.35 | 3.02 [1.6–5.6] | *<0.001 | 1.71 [0.98–3.0] | 0.06 |

*P-value <0.05.

Participants who reported experiencing more violence during the pandemic compared to before had greater odds of reporting heavy alcohol use (OR = 2.07, 95% CI: 1.1–3.8; p = 0.02), a strong desire for alcohol (OR = 2.19; 95% CI: 1.3–3.6; p<0.01) in past month and drinking more during the pandemic (OR = 2.02, 95% CI: 1.2–3.3; p<0.01).

Participants who experienced GBV during the pandemic reported a stronger desire for alcohol (OR = 3.02; 95% CI: 1.6–5.6; p<0.001).

## Multivariate associations between violence and alcohol use

In multivariate models that adjusted for gender, age, race/ethnicity, and HIV status, the significant associations between violence exposure and alcohol remained significant, except for the association between lifetime experience of GBV and strong desire for alcohol in the past month (see Table 3).

## Discussion

Our study examined the impact of violence exposure (verbal, physical, and sexual) before and during the COVID-19 pandemic on alcohol use among adults who drink alcohol. Overall, we found that alcohol-using adults self-reported an increase in violence exposure and alcohol use during the COVID-19 pandemic. Particularly, there were reports of increased violence experiences, gender-based violence, and alcohol use among high proportions of participants. We also found a significant association between violence exposure, including GBV, and elevated alcohol desire and use, in lifetime and during the pandemic. Findings suggest that there is a positive relationship between violence exposure, alcohol use, and the COVID-19 pandemic in which the pandemic compounded the relationship between violence and alcohol use.

Nearly three-fourths of people in our study (72.5%) had a lifetime exposure to verbal, physical, and/or sexual violence, which is higher than comparable measures of lifetime exposure of

Table 3. Multivariate associations of violence exposure and alcohol use (N = 565).

| Violence exposure[a] | Heavy alcohol use AOR [95% CI] | P-value | Strong desire for alcohol AOR [95% CI] | P-value | Drink more during pandemic AOR [95% CI] | P-value |
|---|---|---|---|---|---|---|
| Violence ever | 1.3 [0.8–2.0] | 0.27 | 1.59 [1.1–2.4] | *0.03 | 1.13 [0.7–1.7] | 0.55 |
| GBV ever | 0.85 [0.5–1.4] | 0.51 | 1.33 [0.9–2.1] | 0.29 | 1.12 [0.7–1.8] | 0.62 |
| Violence during pandemic | 1.84 [1.1–3.0] | *0.01 | 2.54 [1.7–3.8] | *<0.01 | 1.54 [1.0–2.3] | *0.04 |
| More violence during pandemic | 2.20 [1.2–4.1] | *0.01 | 2.22 [1.3–3.7] | *0.002 | 2.08 [1.3–3.5] | *0.005 |
| GBV during pandemic | 0.71 [0.4–1.3] | 0.28 | 2.89 [1.5–5.4] | *0.001 | 1.60 [0.9–2.9] | 0.13 |

[a]Each model controls for gender, age, race/ethnicity, and HIV status |

*P-value <0.05.

intimate partner, sexual, and psychological violence in an international study [38]. Generally, it was difficult to find comparable national statistics on lifetime exposure to violence due to a lack of standardization of how lifetime experiences of violence is measured—an issue that other researchers have identified [38]. The closest national estimate of lifetime violence exposure available—an estimate of any exposure to sexual violence, physical violence, and/or stalking by an intimate partner—was 47.3% for women and 44.2% for men [39]. A surveillance study in the SFBA reported that 15.1% of adults surveyed ever experienced physical or sexual violence specifically from an intimate partner [40]. The high percentage of lifetime violence exposure in our study is likely because our estimate combined verbal, physical, and sexual violence in our definition of violence, and perpetrators were not limited to intimate partners. Our study also found that about one-fourth of participants reported experiencing GBV, an estimate that is lower than the prevalence of IPV (34.9%) among women in California. This estimate of lifetime GBV exposure is also lower than an estimate of lifetime violence exposure among trans women (52%) in SFBA due to their gender identity [41]. Our lower GBV estimate is likely because our sample includes cis-gender men.

In terms of alcohol use, three-quarters of people reported heavy alcohol use ($\geq$4 drinks on one occasion $\geq$4 times a month) in the previous three months and about half reported a strong desire for alcohol in the previous month. Our estimates of heavy alcohol use are higher than estimates of heavy alcohol use (6.4%) and binge drinking (23.3%) in the previous month among adults in a 2021 national surveillance study [42]. Our estimates are likely higher than alcohol consumption rates among the general population because the parent study population was people interested in enrolling into an alcohol treatment clinical trial.

The high prevalence of violence experiences in our study suggests that adults who use alcohol and/or are seeking alcohol treatment may have a higher exposure to violence. This was corroborated by our finding that there were significant associations between violence exposure and alcohol use and desire. Specifically, we found that participants who reported a lifetime experience of violence had greater odds of reporting a strong desire for alcohol in the past month. Additionally, those who reported a lifetime experience of GBV had greater odds of reporting a strong desire for alcohol in the past month. These findings reflect those of extant literature on domestic violence and IPV that violence exposure predicts alcohol use [43–45] and that alcohol is used by survivors of violence to cope with trauma symptoms [46,47]. Future longitudinal and qualitative studies are needed to examine the temporal relationship between violence experiences and the context of alcohol use.

In terms of violence exposure and alcohol use during the COVD-19 pandemic, there were increases reported for both, which reflect findings from other studies on alcohol use [7,48] and domestic violence [14] during the pandemic. Specifically, we found that about half of participants reported drinking more alcohol than prior to the pandemic, which is higher than national estimates from a SAMHSA surveillance study of adults who use alcohol (13.4%) in the U.S. [49]. The increase in alcohol use is particularly concerning as studies have found that heavy alcohol use not only worsens mental health outcomes under pandemic conditions [9], but also the severity of COVID-19 infection [10,11].

Additionally, one-third of people surveyed reported experiencing violence during the COVID-19 pandemic and 15.6% reported more violence exposure during the pandemic than usual. A study of California residents two weeks after the shelter-in-place ordinance reported slightly lower violence estimates, with IPV at 15.5% and sexual violence at 10.1% [50]. While some forms of IPV and sexual violence can be categorized as GBV, at the time of our writing, there were no known estimates in the SFBA or nationally of exposure to violence due to one's gender identity or presentation specifically during the pandemic. However, studies elsewhere reported increases in GBV and IPV [14,19,51,52]. The high prevalence of violence during the

pandemic (27.9%) in our study, along with the finding that many of our participants experienced more violence than usual (15.6%), is likely due to an increase in socioeconomic stressors caused by the COVID-19 pandemic and its related changes. As a study on the association between IPV and economic insecurity during the COVID-19 pandemic among women and transgender people in the U.S. found, IPV victimization was worsened by material factors like housing and healthcare insecurity [25]. More generally, studies have found that pandemics and other crises cause or exacerbate social stressors such as job loss and reduced income, which have been found to be drivers of violence [52,53].

An important finding in our study is that there were significant associations between violence exposure and alcohol use during the COVID-19 pandemic. Participants who reported experiencing any violence and those who reported experiencing more violence than usual during the pandemic had greater odds of reporting heavy alcohol use, a strong desire for alcohol in the past month, and of drinking more alcohol during the pandemic. In addition, participants who reported experiencing GBV during the pandemic had greater odds of reporting a strong desire for alcohol in the past month. These findings support reports from other studies in the U.S. and internationally that show a positive association between violence and alcohol use during the pandemic, such as a strengthened relationship between alcohol use at home and domestic violence and of increased IPV and higher alcohol use [30–33]. The linkages between violence exposure and alcohol consumption may potentially be explained by exposure to stressful events which prior studies have linked to increased alcohol use and alcohol use disorders, including COVID-19 specific studies [54–58]. It's plausible that participants in this study may report greater drinking because of the stress associated with the violence they experienced or associated with the pandemic-specific changes. Additional studies may be needed to better understand the linkages we observed.

## Limitations

Our study has limitations. This was a cross-sectional analysis; therefore, we cannot determine whether there's a causal relationship between violence exposure and alcohol use, nor can we ascertain the temporal sequence between the two behaviors. It is also possible that there were potential confounders that we did not adjust for such as job loss or reduced income, which previous studies on COVID-19's impact have found to be associated with alcohol use [6,57,58]. We did not adjust for these potential confounders because our study was cross-sectional and we are not able to rule out whether our measures for these conditions occurred after our exposure and may thereby be part of the causal pathway between our exposure and our outcome. Thus, adjusting for them would potentially introduce collider stratification bias, as well as incorrectly treat a mediator as a confounder [59–61]. Future longitudinal studies with repeated measures might be needed to better account for the role of these potential confounders and confirm the associations we observed. Additionally, our study did not measure participants' alcohol use prior to the pandemic. Hence, we are unable to examine prior study findings that have documented that one's level of alcohol use prior to the COVID-19 pandemic is associated with the volume and frequency of alcohol consumption during the pandemic [62,63]. However, we measured whether participants felt that they drank more during the COVID-19 pandemic. Our analysis found a positive association between being exposed both to violence during the pandemic and more violence during the pandemic and reporting drinking more during the pandemic. Another study limitation is that data was collected in the SFBA and may not be generalizable to other cities. As mentioned, our study was conducted among adults who drink alcohol and were being screened for a clinical trial for AUD treatment. Therefore, the estimates of alcohol use and of violence exposure in our study may be higher

than in the general population. We also did not ask participants to specify the type of violence they experienced (i.e., verbal, physical, or sexual); thus, our estimates of violence experienced may be higher than estimates for specific forms of violence. Finally, since the study was nested under a clinical trial, recruitment spanned over two years, which may introduce chronological bias. Relatedly, data was self-reported and consists of responses to questions that asked participants to recollect experiences that occurred in the past (e.g., history of GBV; experiences during shelter-in-place ordinances). Thus, participant responses may have been prone to social desirability or recall bias.

## Recommendations

Despite these limitations, our study contributes important findings, including pandemic specific data, indicating a positive correlation between violence exposure, GBV, and alcohol use and desire to drink. Drawing on these findings, it is important that social services, both by government and non-governmental organizations, are prepared to provide targeted outreach to support people who are at risk of or experience GBV, as well as those facing increased incidents or severity of violence during pandemics. When appropriate, these services should also screen individuals for AUD with their consent and link them to treatment. Particularly, future pandemic preparedness efforts must develop and incorporate violence prevention strategies, with a focus on the most vulnerable populations, such as women, gender minorities, and people living with substance use disorders. Internationally, about 52 countries incorporated strategies to prevent violence against women and girls into their COVID-19 plans after the first year of the pandemic in light of surging GBV; the U.S. however, had not adopted such a plan [64]. Recommendations like those released by the United Nations Women [17] at the start of the pandemic and review articles highlighting exemplar responses to surging violence and advising an increase in funding, a strengthening of services and organizational capacity, centering marginalized genders in policies, providing social, economic, and virtual support to survivors, and collecting gender disaggregated data to monitor violence, are instructive on future preparedness and responses [16,20,53]. In addition, drawing on their study in Puerto Rico, Gaba and colleagues [20] have suggested that governments use existing global guidance and psychological interventions to address surges of GBV during pandemics—a recommendation that proved effective in addressing surging GBV in Puerto Rico during the COVID-19. Increasing psychological and mental health interventions may mitigate the stress and trauma from violence, and in turn reduce heavy alcohol consumption among those who experienced violence.

Across the U.S. and in the SFBA specifically, organizations that provide services to survivors of domestic, intimate partner, and GBV should allocate resources towards services that address pandemic-related needs of their clients. Such funds can be used to support services such as safe shelter, therapy, and substance use counseling and treatment that can help mitigate the risk of violence and/or substance use to cope with violence trauma. By the same token, state and federal government should provide funds to support efforts to support violence survivors and people with substance use disorders. As domestic violence and GBV advocates and activists in California have pointed out, government funding to support survivors and the organizations that service them is inadequate and are disproportionately allocated to shelter-based programs, which only serve a fraction of survivors, while non-shelter-based programs and culturally responsive programs for underserved communities are underfunded [65]. Additionally, there is no state funding for violence prevention programs, which would help address the issue at its root. Alcohol harm reduction, recovery, and treatment programs should also be adapted to pandemic conditions and made available for those who report violence. Evidence also suggest that jointly addressing both AUD and traumatic stressful events are possible and

efficacious [66]. Likewise, given the link between socioeconomic and psychosocial stressors and violence and alcohol use, social programs that relieve these stressors should be implemented to prevent alcohol use disorder and violence.

## Conclusion

Our study findings contribute data on how COVID-19 pandemic conditions exacerbated social issues, particularly alcohol use and violence exposure and further document the impact of violence on alcohol use. The study also notably contributes to an estimate of GBV exposure during the pandemic, which other studies have measured using the proxy measures of domestic and intimate partner violence. Future efforts in pandemic preparedness must develop and incorporate violence prevention strategies and adapt alcohol harm reduction, recovery, and treatment programs to pandemic conditions. Additionally, more research, including longitudinal studies, should be conducted to determine the impact of specific pandemic stressors on violence and alcohol use.

## Acknowledgments

We would like to express our sincere gratitude to study staff and participants for their time and contributions to the study.

## Author Contributions

**Conceptualization:** Akua O. Gyamerah.

**Data curation:** Akua O. Gyamerah.

**Formal analysis:** Akua O. Gyamerah, Andy C. Canizares, Glenn-Milo Santos.

**Funding acquisition:** Akua O. Gyamerah, Glenn-Milo Santos.

**Investigation:** Akua O. Gyamerah.

**Methodology:** Akua O. Gyamerah, Willi McFarland, Erin C. Wilson, Glenn-Milo Santos.

**Project administration:** Akua O. Gyamerah, Janet Ikeda, Glenn-Milo Santos.

**Supervision:** Akua O. Gyamerah, Janet Ikeda, Glenn-Milo Santos.

**Writing – original draft:** Akua O. Gyamerah, Alexandrea E. Dunham.

**Writing – review & editing:** Alexandrea E. Dunham, Janet Ikeda, Andy C. Canizares, Willi McFarland, Erin C. Wilson, Glenn-Milo Santos.

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
