## [Decision Letter · Decision Letter 0]

16 Oct 2024

PONE-D-24-21093Impact of the COVID-19 pandemic on violence exposure and alcohol use among adults who drink alcoholPLOS ONE

Dear Dr. Gyamerah,

Thank you for submitting your manuscript to PLOS ONE. After careful consideration, we feel that it has merit but does not fully meet PLOS ONE’s publication criteria as it currently stands. Therefore, we invite you to submit a revised version of the manuscript that addresses the points raised during the review process.

**ACADEMIC EDITOR: **

I inform you that we are unable to accept your manuscript for publication in its current form.

The reviewers have provided valuable feedback on your manuscript, with the two reviews being positive, but some changes to the manuscript are recommended. Therefore, we recommend that you thoroughly address all the comments provided by the reviewers and revise your  manuscript accordingly.

It is essential that you address these concerns comprehensively in your revised manuscript to ensure its suitability for publication.

We understand that revising your manuscript may require substantial effort, but we believe that addressing the reviewers' comments will significantly improve the quality and impact of your research. We encourage you to carefully consider all the feedback provided and to make the necessary revisions accordingly.

Please note that your revision may be subject to further review and that this initial decision does not guarantee acceptance.

We look forward to receiving your revised manuscript.

Kind regards,

Claudio Alberto Dávila-Cervantes, Ph.D.

Academic Editor

PLOS ONE

Journal Requirements:

1. When submitting your revision, we need you to address these additional requirements. Please ensure that your manuscript meets PLOS ONE's style requirements, including those for file naming. The PLOS ONE style templates can be found at https://journals.plos.org/plosone/s/file?id=wjVg/PLOSOne_formatting_sample_main_body.pdf and https://journals.plos.org/plosone/s/file?id=ba62/PLOSOne_formatting_sample_title_authors_affiliations.pdf 2. Thank you for stating the following financial disclosure: "This research was supported by funds from a National Institute of Alcohol Abuse and Alcoholism (NIAAA) Diversity Supplement (3R01AA025930-03S1; parent study: R01AA025930-01A1; PI: Glenn-Milo Santos, PhD)." Please state what role the funders took in the study.  If the funders had no role, please state: "The funders had no role in study design, data collection and analysis, decision to publish, or preparation of the manuscript." If this statement is not correct you must amend it as needed. Please include this amended Role of Funder statement in your cover letter; we will change the online submission form on your behalf.  3. Thank you for uploading your study's underlying data set. Unfortunately, the repository you have noted in your Data Availability statement does not qualify as an acceptable data repository according to PLOS's standards. At this time, please upload the minimal data set necessary to replicate your study's findings to a stable, public repository (such as figshare or Dryad) and provide us with the relevant URLs, DOIs, or accession numbers that may be used to access these data. For a list of recommended repositories and additional information on PLOS standards for data deposition, please see https://journals.plos.org/plosone/s/recommended-repositories. 4. Your ethics statement should only appear in the Methods section of your manuscript. If your ethics statement is written in any section besides the Methods, please move it to the Methods section and delete it from any other section. Please ensure that your ethics statement is included in your manuscript, as the ethics statement entered into the online submission form will not be published alongside your manuscript.

Reviewers' comments:

Reviewer's Responses to Questions

**Comments to the Author**

1. Is the manuscript technically sound, and do the data support the conclusions?

Reviewer #1: Partly

Reviewer #2: Yes

2. Has the statistical analysis been performed appropriately and rigorously? 

Reviewer #1: Yes

Reviewer #2: Yes

3. Have the authors made all data underlying the findings in their manuscript fully available?

Reviewer #1: Yes

Reviewer #2: Yes

4. Is the manuscript presented in an intelligible fashion and written in standard English?

Reviewer #1: No

Reviewer #2: Yes

5. Review Comments to the Author

Reviewer #1: The analysis presented in the paper adds significant insights on the linkage between violence and alcohol consumption during the COVID-19 pandemic in a selective urban population. The study, however, has significant limitations mainly related to the unmeasured confounders, which shall me discussed in the discussion section and mentioned in the study limitations subsection. Violence is only one among several important causes of pandemic associated distress.

The fact is that violence is just one of several important causes of pandemic-related distress that may have influenced alcohol consumption. Quite a lot of research works has already been published in the scientific literature that show the relationship between other causes of distress during the pandemic that influenced alcohol consumption. They included job loss during the pandemic, decreased income during the pandemic, and the perception of the pandemic itself as a stressful event.

Also, according to the published literature, alcohol consumption and levels of violence were influenced by the strength of the restrictions imposed to reduce spread of COVID-19 and the severity of self-isolation controls during the pandemic. In the discussion, authors are encouraged to provide some details about the severity of restrictions during the pandemic in the community surveyed. The authors should also mention that the study did not ask questions about the respondent’s own perception of the severity of the imposed restrictions on social and everyday life, which, according to the literature, may affect both the level of violence and the level of alcohol consumption.

Several studies reported association between the levels of alcohol consumption before pandemic and increase in alcohol consumption during the pandemic.

For example, one study conducted among users of social media demonstrated a positive association between the increase in the frequency of alcohol consumption and severe restrictions in everyday private life (OR: 3.127; 95% CI: 1.011–9.675), and severe negative professional or financial consequences due to the spread of SARS-CoV-2 (OR: 2.247; 95% CI: 1.131–4.465). In this study, the odds of an increase in the frequency of heavy episodic drinking were more than twice higher (OR: 2.329; 95% CI: 1.001–5.428) among those who had experienced severe negative consequences of the pandemic to their professional and financial situation. Higher typical frequency and usual consumption (volume) of alcohol on a typical drinking occasion and higher typical frequency of heavy episodic drinking before the pandemic were positively significantly associated with the increase in alcohol consumption during the pandemic. (Reference: Gil AU, Demin AK. Factors associated with increase in alcohol consumption during first months of COVID-19 pandemic among online social media users in Russia. Bulletin of Russian State Medical University. 2021. 6:(118-128). DOI 10.24075/brsmu.2021.064.). Authors shall also discuss in the discussion section the literature dedicated to the polarization of alcohol consumption during the pandemic where it was found that heavy drinkers tend to increase consumption of alcohol during the pandemic, while those drinking less tend to increase alcohol consumption less or to decrease their alcohol consumption. Initial (before pandemic) level of alcohol consumption (frequency, one-time volume of drinking, frequency of heavy episodic drinking) is an important unmeasured confounder in analysis presented in the submitted paper.

Among published works on the mentioned above unmeasured confounders, the following are recommended for discussion in the discussion section of the paper:

Rossow I, Bartak M, Bloomfield K, et al. Changes in Alcohol Consumption during the COVID-19 Pandemic Are Dependent on Initial Consumption Level: Findings from Eight European Countries. Int J Environ Res Public Health. 2021;18(19):10547.

Changes in self-reported alcohol consumption at high and low consumption in the wake of the COVID-19 pandemic: A test of the polarization hypothesis

Alexander Tran, et al. https://www.medrxiv.org/content/10.1101/2024.07.31.24311291v1

Kilian, C., Rehm, J., Allebeck, P., Braddick, F., Gual, A., Barták, M., . . . O'Donnell, A. (2021). Alcohol

consumption during the COVID-19 pandemic in Europe: a large-scale cross-sectional study in 21

countries. Addiction, 116(12), 3369-3380.

Gil AU, Demin AK. Factors associated with increase in alcohol consumption during first months of COVID-19 pandemic among online social media users in Russia. Bulletin of Russian State Medical University. 2021. 6:(118-128). DOI 10.24075/brsmu.2021.064. https://cyberleninka.ru/article/n/factors-associated-with-increase-in-alcohol-consumption-during-first-months-of-covid-19-pandemic-among-online-social-media-users-in/viewer

Gil A., et al. Changes in alcohol consumption in the Russian Federation during the first months of the COVID-19 pandemic. https://cyberleninka.ru/article/n/izmeneniya-osobennostey-potrebleniya-alkogolya-v-rossiyskoy-federatsii-v-pervye-mesyatsy-pandemii-covid-19/viewer

and other studies available in the reference lists of indicated above publications.

Once, the issue of unmeasured confounders is thoroughly discussed in the paper, the revised manuscript can be considered for publication.

Reviewer #2: Thank you for the opportunity to review this article examining the association between violence (and GBV) and alcohol use during the COVID-19 pandemic in the San Francisco Bay Area. The article is well written and provides important insight into this under-researched topic. Building on the findings, the authors also make strong recommendations relevant to this discipline and well-placed within the context of previous literature.

I am of the opinion that this article should be accepted for publication, with only minor revisions required to provide additional clarity for the reader.

Minor revisions:

1. Page 4, Introduction: "Alcohol purchases increased by 54% in the first week of shelter-in-place orders in the US (5)." This seems to reference a journal article that referenced this statistic in the introduction and not the source material. Please ensure all citations cite the appropriate reference and original source material for the reader. Please also provide additional information on what the 54% increase in sales is being compared to (e.g. the previous year) and if the data is based on sales or self-reported purchasing.

2. Page 6, Introduction: "In the United States, a study using 911 calls in Michigan documented a relationship between domestic violence reports and alcohol use (30)" please include a brief description of the nature of the relationship/association.

3. Page 6, Introduction: The first sentence in the aims paragraph took a few reads. Please revise for clarity. Perhaps consider splitting over two sentences, and/or utilise additional commas.

4. Methods and Materials: Please advise how interested participants responded to the recruitment drive (e.g. did they have to call, text, email etc). Would the required method of responding to the recruitment advertisements have affected who responded?

5. Methods and Materials: Please include whether you have followed any reporting guidelines or checklists. If not, please consider applying one like STROBE and ensure you have included all relevant information for reporting. e.g. was there any missing data? How was it managed?

6. Page 10, results: "Nearly one-third (27.9%) of participants" this is closer to a quarter, a third could be considered overstating the results. Consider revising two 'over a quarter' or something similar.

7. Page 12, Discussion: "Overall, we found that the COVID-19 pandemic increased violence exposure and alcohol use among alcohol-using adults." consider revising to more accurately reflect the findings, e.g. "...alcohol-using adults self-reported an increase in violence exposure and alcohol use during the covid-19 pandemic"

8. Discussion: Please consider swapping the order of paragraphs 2 and 3 of the discussion so it flows more naturally from discussing why the rates of violence identified in your study are higher than that of the general population, and that this high prevalence of alcohol use is likely related to the population you are looking at being adults who use alcohol and/or seeking alcohol treatment. As it is written now, it felt like this was an overlooked point in paragraph 2.

Given the length of the discussion, please also consider the use of subheadings if this falls within journal guidelines - perhaps a section for covid-related findings.

9. Page 14, Discussion: "half of participants reported drinking more alcohol than prior to the pandemic, which is higher than national estimates from the same surveillance study of adults who use alcohol (13.4%) in the U.S. (49)." unclear what is meant by 'the same surveillance study' here, please revise to clarify

10. Page 15, Discussion: "The high prevalence of violence during the pandemic in our study along with the finding that many experienced"; please revise to something like 'many of our participants' or something similar for clarity. It would also be beneficial to include the percentage in brackets here to save the reader having to refer back.

11. Page 16, recommendations: "be prepared to provide targeted outreach to support people who are at risk of or experience GBV and elevated violence" please add more clarity here for if you mean people who have an elevated risk of violence and GVB, or describe what you mean by 'elevated violence'.

12. Please also advise when the shelter in place order(s) for the SFBA ceased in comparison to when the recruitment drive stopped, and include any commentary on if you think this may have affected the results.

Minor typographical or grammar suggestions:

- Page 2, Abstract, 6th line from bottom: replace 'thank' with 'than'

- Pandemic and epidemic have been used seemingly interchangeably. Advise all be revised to pandemic if referring to COVID-19, or clearly include why epidemic is being used.

- Page 5, Introduction: "GBV can consist of violence experienced from strangers, that state (i.e., the police, military, etc.)," revise to 'the' state

- Page 5+6, Introduction: IPV acronym introduced twice. Suggest it instead be introduced in sentence "with a few national and local studies reporting significant increases in violence experiences, especially IPV, among cis women and gender minorities" and remove from other instances

- Page 5, Introduction: double opening bracket on reference 19

- Page 5, Introduction: Full stop/period missing from sentence ending "similarly found that the pandemic worsened incidence of domestic violence against women"

- Page 5, Introduction: In sentence following the above, revise tense for "reporting"

- Page 14, Discussion: "While some forms of IPV and sexual violence can be categorized at GBV"; 'as' instead of 'at'

- Page 16, Limitations: Please revise the fourth sentence in this paragraph

6. PLOS authors have the option to publish the peer review history of their article (what does this mean?). If published, this will include your full peer review and any attached files.

Reviewer #1: **Yes: **Artyom Gil

Reviewer #2: No

---

## [Author Response · Author response to Decision Letter 0]

3 Dec 2024

Dear Dr. Dávila-Cervantes,

Thank you for the opportunity to revise and resubmit our manuscript. We have given careful consideration to all the issues raised by reviewers and have revised our manuscript accordingly. Below is a point-by-point clarification and explanation of revisions made in response to the reviewers. We believe our revision address all concerns raised and is suitable to publication now. Our sincere gratitude to you and our manuscript reviewers for the insightful feedback.

Sincerely,

Manuscript authors

Reviewer comments Authors’ responses

Journal requirements

1. When submitting your revision, we need you to address these additional requirements. Please ensure that your manuscript meets PLOS ONE's style requirements, including those for file naming.

Response: Thank you for this suggestion. We have reviewed the journal requirements and ensured that our manuscript meets all of PLOS ONE’s style requirements.

2. Thank you for stating the following financial disclosure: "This research was supported by funds from a National Institute of Alcohol Abuse and Alcoholism (NIAAA) Diversity Supplement (3R01AA025930-03S1; parent study: R01AA025930-01A1; PI: Glenn-Milo Santos, PhD)."

Response: Thank you for flagging this. We have added the following statement to our financial disclosure on the second page of our cover letter.

3. Thank you for uploading your study's underlying data set. Unfortunately, the repository you have noted in your Data Availability statement does not qualify as an acceptable data repository according to PLOS's standards.

Response: We apologize for sharing the wrong repository link. We have updated the link on the submission form for your reference. We have also posted it below.

http://datadryad.org/stash/share/RoK9itq9yFqVio-7R7M5hqRIL87rsMJ8D-bEgjjax5M

Response: Thank you for the correction. We have removed all other references to the ethics statement and have left only one reference to it in the Methods section.

Response: We have reviewed our reference list to ensure it is complete. We have also added six more references and removed one reference in response to reviewer comments:

Removed reference:

5. Pollard MS, Tucker JS, Green HDJ. Changes in Adult Alcohol Use and Consequences During the COVID-19 Pandemic in the US. JAMA Netw Open. 2020 Sep 1;3(9):e2022942.

Added references:

5. Castaldelli-Maia JM, Segura LE, Martins SS. The concerning increasing trend of alcohol beverage sales in the U.S. during the COVID-19 pandemic. Alcohol Fayettev N. 2021 Nov;96:37–42.

57. Acuff SF, Strickland JC, Tucker JA, Murphy JG. Changes in alcohol use during COVID-19 and associations with contextual and individual difference variables: A systematic review and meta-analysis. Psychol Addict Behav. 2022 Feb;36(1):1–19. 

58. Weerakoon SM, Jetelina KK, Knell G, Messiah SE. COVID-19 related employment change is associated with increased alcohol consumption during the pandemic. Am J Drug Alcohol Abuse. 2021 Nov 2;47(6):730–6. 

59. Groenwold RHH, Palmer TM, Tilling K. To Adjust or Not to Adjust? When a “Confounder” Is Only Measured After Exposure. Epidemiol Camb Mass. 2021 Mar 1;32(2):194–201. 

60. Greenland S. Quantifying biases in causal models: classical confounding vs collider-stratification bias. Epidemiol Camb Mass. 2003 May;14(3):300–6. 

61. Ananth CV, Schisterman EF. Confounding, Causality and Confusion: The Role of Intermediate Variables in Interpreting Observational Studies in Obstetrics. Am J Obstet Gynecol. 2017 Aug;217(2):167–75. 

62. Gil AU, Demin AK. Factors associated with increase in alcohol consumption during first months of COVID‑19 pandemic among online social media users in Russia. Bull Russ State Med Univ. 2021;(6):118–28. 

63. Rossow I, Bartak M, Bloomfield K, Braddick F, Bye EK, Kilian C, et al. Changes in Alcohol Consumption during the COVID-19 Pandemic Are Dependent on Initial Consumption Level: Findings from Eight European Countries. Int J Environ Res Public Health. 2021 Oct 8;18(19):10547.

6. Pandemic and epidemic have been used seemingly interchangeably. Advise all be revised to pandemic if referring to COVID-19, or clearly include why epidemic is being used. Thank you for this note. We have taken your advice to revise “epidemic” to “pandemic” in order to be consistent throughout the paper.

Response to Reviewer 1 Comments

7. The analysis presented in the paper adds significant insights on the linkage between violence and alcohol consumption during the COVID-19 pandemic in a selective urban population. The study, however, has significant limitations mainly related to the unmeasured confounders, which shall me discussed in the discussion section and mentioned in the study limitations subsection. Violence is only one among several important causes of pandemic associated distress.

The fact is that violence is just one of several important causes of pandemic-related distress that may have influenced alcohol consumption. Quite a lot of research works has already been published in the scientific literature that show the relationship between other causes of distress during the pandemic that influenced alcohol consumption. They included job loss during the pandemic, decreased income during the pandemic, and the perception of the pandemic itself as a stressful event.

Also, according to the published literature, alcohol consumption and levels of violence were influenced by the strength of the restrictions imposed to reduce spread of COVID-19 and the severity of self-isolation controls during the pandemic. In the discussion, authors are encouraged to provide some details about the severity of restrictions during the pandemic in the community surveyed. 

The authors should also mention that the study did not ask questions about the respondent’s own perception of the severity of the imposed restrictions on social and everyday life, which, according to the literature, may affect both the level of violence and the level of alcohol consumption.

Several studies reported association between the levels of alcohol consumption before pandemic and increase in alcohol consumption during the pandemic.

Higher typical frequency and usual consumption (volume) of alcohol on a typical drinking occasion and higher typical frequency of heavy episodic drinking before the pandemic were positively significantly associated with the increase in alcohol consumption during the pandemic. (Reference: Gil AU, Demin AK. Factors associated with increase in alcohol consumption during first months of COVID-19 pandemic among online social media users in Russia. Bulletin of Russian State Medical University. 2021. 6:(118-128). DOI 10.24075/brsmu.2021.064.). 

Authors shall also discuss in the discussion section the literature dedicated to the polarization of alcohol consumption during the pandemic where it was found that heavy drinkers tend to increase consumption of alcohol during the pandemic, while those drinking less tend to increase alcohol consumption less or to decrease their alcohol consumption. Initial (before pandemic) level of alcohol consumption (frequency, one-time volume of drinking, frequency of heavy episodic drinking) is an important unmeasured confounder in analysis presented in the submitted paper. 

Response: Thank you for your insightful and resourceful feedback. You raise an important concern about the potential role of unmeasured confounders in our analysis such as job or income loss. 

First, regarding potential confounders, we did not adjust for those covariates because it is possible that they occurred after our exposure and that they may be part of the causal pathway between our exposure and our outcome and doing so would potentially introduce collider stratification bias as well as potentially treat a mediator as a confounder. 

We acknowledge that there are limitations to not conducting a confounder analysis. We discuss these limitations under the Limitations section and cite relevant sources to support our reasoning on page 15.

Below, we cite an excerpt from an article that elaborates on our reasoning for not adjusting for these covariates.

https://pubmed.ncbi.nlm.nih.gov/33470711/which has citations you can pull from.

 “Textbooks on epidemiology, as well as research articles about confounding adjustment, generally advise against controlling for variables that are measured after exposure has started. One reason is that adjustment for postexposure variables may lead to collider stratification bias. Another reason is that such variables may actually be mediators of the causal relationship between exposure and outcome, and conditioning on such variables may introduce bias. As indicated by VanderWeele: “…we often refrain from adjusting for covariates that occur temporally subsequent to the exposure.”

Second, our discussion including our limitations section now include an expanded discussion that explains that other stressors that have been found to be associated with increased alcohol use might be mediating or potentially confounding the relationship between violence exposure and alcohol use on pages 14-15.

Finally, the findings that higher consumption of alcohol use prior to the pandemic was positively associated with increased alcohol consumption and that those who drank less ten to have less increased consumption are interesting. Our study did not measure participants’ alcohol use prior to the pandemic and thus could not conduct a confounder analysis. However, we measured whether participants felt that they drink more during the COVID-19 pandemic than prior to the pandemic. Our analysis found that those who reported experiencing violence and those who reported experiencing more violence reported drinking more during the pandemic. We have added this discussion to the limitations section as well on page 15.

Response to Reviewer 2 Comments

Abstract

8. Page 2, Abstract, 6th line from bottom: replace 'thank' with 'than' 

Response: Thank you for noticing this error. We have corrected the typo on page 2. It now states “than”.

Introduction

9. Page 5, Introduction: "GBV can consist of violence experienced from strangers, that state (i.e., the police, military, etc.)," revise to 'the' state

Response: We appreciate your note and have updated “that” to say “the.”

10. Page 5+6, Introduction: IPV acronym introduced twice. Suggest it instead be introduced in sentence "with a few national and local studies reporting significant increases in violence experiences, especially IPV, among cis women and gender minorities" and remove from other instances

Response: Thank you for this suggestion. We now recognize that “IPV” as an acronym was introduced twice and have corrected it to only be introduced the first time it appears in the paper. This can be found on page 6 in the sentence: “GBV can consist of violence experienced from strangers, the state (i.e., the police, military, etc.), or one’s family, community, or intimate partner (IPV).”

11. Page 5, Introduction: double opening bracket on reference 19

Response: We have deleted one of the opening brackets and appreciate your note.

12. Page 5, Introduction: Full stop/period missing from sentence ending "similarly found that the pandemic worsened incidence of domestic violence against women" 

Response: Thank you. We have added a period to the end of this sentence.

13. Page 5, Introduction: In sentence following the above, revise tense for "reporting" 

Response: Thank you. We have changed the tense of “reporting” to “reported.” 

14. Page 6, Introduction: The first sentence in the aims paragraph took a few reads. Please revise for clarity. Perhaps consider splitting over two sentences, and/or utilise additional commas. 

Response: Thank you for the suggestion. We have revised this sentence by removing some of the redundancies in word choice, thus increasing the overall clarity of the sentence. Here is the revised version: 

“While the impact of the pandemic on alcohol use and on GBV has been documented, few studies have examined the co-occurrence between the two—an issue of concern given prior research demonstrating a relationship between alcohol use and violence.”

15. Page 4, Introduction: "Alcohol purchases increased by 54% in the first week of shelter-in-place orders in the US (5)." This seems to reference a journal article that referenced this statistic in the introduction and not the source material. Please ensure all citations cite the appropriate reference and original source material for the reader. Please also provide additional information on what the 54% increase in sales is being compared to (e.g. the previous year) and if the data is based on sales or self-reported purchasing. 

Response: Thank you for this comment. We referred back to the original source and could not find the exact percentage. We have thus changed the reference to another source and have modified the sentence to the following on page 3:

“Alcohol sales in the U.S. increased by 20% in the first six months of the pandemic (March 2020 and September 2020) in comparison to sales during the same time period in 2019 (5).” 

16. Page 6, Introduction: "In the United States, a study using 911 calls in Michigan documented a relationship between domestic violence reports and alcohol use (30)" please include a brief description of the nature of the relationship/association. 

Response: Thank you for this feedback! We have refined the statement explaining this study to better describe the relationship/association. See as follows:

“In the United States, a study using 911 calls in Michigan documented that the strength of the relationship between domestic violence reports and alcohol sales more than doubled since the start of the pandemic.”

Methods

17. Methods and Materials: Please advise how interested participants responded to the recruitment drive (e.g. did they have to call, text, email etc). Would the required method of responding to the recruitment advertisements have affected who responded? 

Response: Thank you for suggesting that we clarify this point. We have added the following sentence to share how participants were contacted: “To ensure multiple modes of contact, interested parties could contact o

---

## [Editor Report · Decision Letter 1]

6 Dec 2024

Impact of the COVID-19 pandemic on violence exposure and alcohol use among adults who drink alcohol

PONE-D-24-21093R1

Dear Dr. Gyamerah,

We’re pleased to inform you that your manuscript has been judged scientifically suitable for publication and will be formally accepted for publication once it meets all outstanding technical requirements.

Kind regards,

Claudio Alberto Dávila-Cervantes, Ph.D.

Academic Editor

PLOS ONE
---

## [Editor Report · Acceptance letter]

18 Dec 2024

PONE-D-24-21093R1 

PLOS ONE

Dear Dr. Gyamerah, 

I'm pleased to inform you that your manuscript has been deemed suitable for publication in PLOS ONE. Congratulations! Your manuscript is now being handed over to our production team.

Kind regards, 

on behalf of

Mr. Claudio Alberto Dávila-Cervantes 

Academic Editor

PLOS ONE